# Whole-Body Vibration Associated with Strength Training on the Lower-Limb Blood Flow and Mobility in Older Adults with Type 2 Diabetes: A Study Protocol for a Randomized Controlled Trial

**DOI:** 10.3390/diagnostics12071550

**Published:** 2022-06-25

**Authors:** François Talles Medeiros Rodrigues, Ana Paula de Lima Ferreira, Kennedy Freitas Pereira Alves, Thais Vitorino Marques, Daniel Florentino de Lima, Larissa Coutinho de Lucena, Shirley Lima Campos, Wagner Souza Leite, Ricardo Oliveira Guerra, Amandine Rapin, Maria das Graças Rodrigues de Araújo, Redha Taiar

**Affiliations:** 1Laboratório de Cinesioterapia e Recursos Terapêuticos Manuais (LACIRTEM), Departamento de Fisioterapia, Universidade Federal de Pernambuco (UFPE), Recife 50670-901, PE, Brazil; francoismedeirosfisiot@gmail.com (F.T.M.R.); apllima@yahoo.com.br (A.P.d.L.F.); kennedyfpa@hotmail.com (K.F.P.A.); thaisvmarques@hotmail.com (T.V.M.); daniel.florentino.lima@gmail.com (D.F.d.L.); mgrodriguesaraujo@hotmail.com (M.d.G.R.d.A.); 2Faculdade Nova Esperança (FACENE), João Pessoa 58067-698, PB, Brazil; larissacoutinho@gmail.com; 3Laboratório Multiusuário de Inovação Instrumental e Desempenho Físico-Funcional (LInDEF), Departamento de Fisioterapia, Universidade Federal de Pernambuco (UFPE), Recife 50670-901, PE, Brazil; shirleylcampos@uol.com.br (S.L.C.); wagnerszleite@gmail.com (W.S.L.); 4Grupo de Estudos em Epidemiologia e Fisioterapia Geriátrica (GEFEG), Departamento de Fisioterapia, Universidade Federal do Rio Grande do Norte (UFRN), Natal 59078-970, RN, Brazil; ricardoguerra2009@gmail.com; 5Faculté de Médecine, Université de Reims Champagne Ardennes, UR 3797 VieFra, 51097 Reims, France; arapin@chu-reims.fr; 6MATériaux et Ingénierie Mécanique (MATIM), Université de Reims Champagne Ardenne, 51100 Reims, France

**Keywords:** ClinicalTrial.gov, NCT03443986, diabetes type 2, vascular endothelium, peripheral nervous system, whole-body vibration, strength training

## Abstract

Vascular endothelium insults caused by high serum glucose levels affect the oxygen supply to tissues, via the microvascular endothelium, resulting in an increased perfusion heterogeneity. These insults may lead to the underuse of blood capillaries, while other vessels are overused and effectively overload their oxygen supply capacity, which eventually causes damages to distal parts of the peripheral nervous system. Therefore, the proprioceptive and exteroceptive feedback information will be gradually lost and contribute to a mobility reduction. This study aims to assess the efficacy of whole-body vibration (WBV) associated with strength training (ST) on lower-limb blood flow and mobility in older adults with type 2 diabetes (DM2). Methods and analyses: This is a protocol (1st version) for Pa single-blind, randomized, controlled clinical trial guided by the SPIRIT guidelines. Our sample will consist of 51 older adults with DM2 randomly allocated to three groups: low frequency WBV (16–26 Hz) associated to ST (G1), WBV sham (G2) and nonintervention control (G3). The study protocol is set for a 12-week (three times per week) schedule. Primary outcomes: skin temperature using infrared thermographic imaging (ITI); mean peripheral arterial blood flow velocity (MBF) by a handheld Doppler ultrasound (DU), and functional mobility by Timed Up and Go (TUG) test. Secondary outcomes: quasi-static posture using the DX100 BTS Smart optoelectronic system, and plantar pressure and body balance using the MPS stabilometric platform. Data will be collected and analyzed at baseline and post-intervention, considering *p*-value < 0.05 level of significance. The analyses will also be conducted with an intention-to-treat method and effect size. Dissemination: All results will be published in peer-reviewed journals as well as presented in conferences.

## 1. Introduction

The natural course of aging is characterized by a progressive decline in multi-organ function that entails metabolic changes and sarcopenia [1]. However, under stress (physical and/or mental), these changes can affect homeostatic mechanisms and organic response, which decreases the capacity for storage, defense, and adaptation [2]. This imbalance generated by increased levels of free radicals and decreased antioxidant potential (deficiency of vitamins and other antioxidants) causes structural and functional oxidative damage to macromolecules, thus making them susceptible to develop pathological processes. [3,4] Among these processes, diabetes mellitus (DM) stands out for its high rate of morbidity and mortality, as well as for its association with the increase in polypharmacy and its risks of interaction, common in the elderly [5,6].

DM is an emerging global public health issue of the 21st century [7]. It is estimated that 425 million adults are affected by this disease worldwide; 12.5 million of them in Brazil [8], resulting in high social and economic burdens. Healthcare costs related to DMI were estimated between 1.197 and 6.73 billion USD worldwide and 22 billion USD in Brazil in 2015 [9].

DM involves a group of metabolic disorders marked by high blood glucose levels resulting from insufficient insulin production and/or action [10]. Among the several clinical phenotypes, type 2 diabetes (DM2) accounts for the most cases (90 to 95%) [9] and there appears to be a higher incidence in older men [11].

Chronic hyperglycemia can result in damage, dysfunction, and multi-organ and system failures, mainly, related to microcirculation [5]. Functional and structure changes in the vascular endothelium compromise perfusion regulation and consequently, may prejudice the oxygen supply to peripheral tissues [12]. This heterogeneity in perfusion is due to under-utilization of a few capillaries, whereas other capillaries are overused and thus overload [13]. 

Additionally, sub-perfusion and tissue hypoxia generate progressive damage to somatic nerve fibers (sensitive and motor) and to autonomous nervous system [14], which may cause posture instability [15], loss of balance [16], limited mobility [17], and consequently poor quality of life. Therefore, the assessment of regional or specific early perfusion is presented as an important predictor of cardiovascular diseases and can be performed noninvasively using infrared thermography and Doppler ultrasound [7].

Evidence has shown that regular physical activity is an important ally for the endothelium function recovery [18]. Among the physical activity alternatives, it is well established that strength training (ST) has several benefits for the vascular function, since muscle contraction stimulates an increase in glucose uptake by muscle cells [19].

The whole-body vibration (WBV) training program is an alternative for a safe, effective, and low-cost exercise, which can present a satisfying compliance by previously sedentary patients [20,21]. It is generally assumed that vibration stimulates muscle spindles that responds with tonic vibration reflexes thereby enhancing muscle activity [21,22,23]. In the literature, two studies identified a significant blood flow increase in DM2 patients undergone vibration [24,25]. Moreover, two systematic reviews showed positive outcomes related to WBV on glycemic control and other risks associated with diabetes [26,27].

On the other hand, when WBV is associated with ST, a few studies reported substantial positive effect on musculoskeletal response in older women [28] and in untrained adults [29,30]. In the meantime, the evidence for possible additional benefits from this combination for the recovery of endothelial function in diabetics are scarce.

Based on the above considerations, this paper presents a detailed protocol for a single-blind, randomized, controlled clinical trial guided by the SPIRIT (Standard Protocol Items: Recommendations for Interventional Trials) guidelines [31], which aims to identify additional effects of WBV associated to ST. The results will follow CONSORT (Consolidated Standards of Reporting Trials) recommendations for nonpharmacological interventions [32]. The objectives of this study are, primarily, to assess the efficiency of WBV associated with ST on lower-limb blood flow and mobility of older adults with DM2 and, secondly, to verify responses to semi-static position, plantar position and stance. 

Our hypothesis is that the addition of WBV to ST enhances benefits in lower-limb blood flow and mobility of older adults with DM2.

## 2. Materials and Methods

This randomized clinical trial was approved by ethics committee of the research institution, Universidade Federal de Pernambuco, under No. 2.449.800, and it complies with the ethical aspects based on Resolution 466/12 of the National Health Council and the Declaration of Helsinki. 

All volunteers will provide written consent after being informed of the study procedures, possible risks and their right to withdraw at any time, with no consequences for any further treatment. The confidentiality of volunteer data will be protected.

According to the recommendations of the CONSORT extension, possible damages will be reported, resulting in the immediate interruption of treatment [33]. The volunteer will be referred to the emergency department of the research institution hospital if a serious adverse event to the treatment is identified, such as functional disability and/or a life threatening outcome.

The dataset will be stored in a locked room in the LACIRTEM lab in password-protected computer files. These data will be disseminated after completion of the study in international peer-reviewed scientific journals and presented at conferences. Positive, negative and inconclusive results will be published in peer-reviewed journals and presented at national and international conferences. In addition, the data will be shared with the volunteers.

### 2.1. Experimental Design and Location

This is a 12-week (3 times a week), single-blind, randomized, controlled clinical trial with three arms (low-frequency WBV associated with ST, G1; WBV sham associated with ST, G1; and control, G3). The study will be developed in the Kinesiotherapy and Manual therapy Laboratory (LACIRTEM) of the Department of Physiotherapy in the Universidade Federal de Pernambuco (UFPE). Figure 1 is a detailed flow chart of the volunteers.

### 2.2. Sampling and Enrollment Strategies

Older adults with DM2 will be selected from a teaching clinic of physiotherapy waiting list, outpatients physical therapy department of a teaching hospital (Hospital das Clínicas—HC), Nucleo de Atenção ao Idoso (NAI) a healthcare center for old people, and Universidade Aberta à Terceira Idade (UNATI), an education service for old people. The sampling will be performed by convenience. Enrollment strategies will include social media advertisements and media relation communication service (Assessoria de comunicação social—ASCOM/UFPE), as well as handouts distributions and lectures. 

In response to first contact, the volunteers will be invited for a full anamnesis and physical exam with an evaluator to assess their eligibility for the study.

### 2.3. Eligibility Criteria

Inclusion: Men and women between 60 and 80 years old with 2 years of clinical diagnosis of DM2 and pre-obesity (25.0–29.9 kg/m^2^) or with obesity class I (30.0–34.9 kg/m^2^) [34], functionally independent, with no severe foot deformity that may require special footwear, other orthopedic deformities, no high risk of falls, no balance disorders, deep venous thrombosis or need for assistive devices for ambulation, preserved cognitive function assessed by Mini-Mental State Examination (MMSE), considering 19 as cutoff point for poor cognitive function [35], and sedentary (0 to 5 points) or poorly engaged in physical active (6 to 11 points) accordingly to the questionnaire of Habitual Physical Activity (HPA) [36].

Exclusion: Change in the subject’s current medication during the study, secondary physical activity during the training program and persistent hypertensive crisis (≥180 × 110 mmHg) or elevated capillary blood glucose (≥300 mg/dL) that prevent the training program completion.

No compliance: Attendance rate lower than 75% of the scheduled sessions, corresponding to 27 sessions. 

### 2.4. Sample Size

The sample size was based on the primary outcome of lower-limb blood flow circulation evaluated by infrared thermographic imaging (ITI). A minimal clinically important difference for skin temperature asymmetry on the knee anterior region is over 0.5 °C [37]. Based on data from a pilot study, two proportions were compared (G1 = 40% and G3 = 10%), considering power of 80% to detect differences between groups and 5% level of significance. The following formula was used:n = [(Z*α*/2 + Zβ)^2^ P(1 − P)]/(P − Po)^2^

According to the calculation in [38], a sample of approximately 15 volunteers per arm would be necessary. We expect a realistic drop-out rate of about 20%; therefore, 17 volunteers are estimated to be enrolled in each group (total of 51 volunteers).

### 2.5. Randomization and Blinding

The volunteers will be randomly assigned to one of the three arms: low-frequency WBV (16–26 Hz) associated with ST (G1), WBV sham associated with ST (G2), and an untreated group (G3). A researcher not involved in the study will perform the volunteers’ randomization using the webpage www.randomization.com (accessed on 20 January 2018). and subsequently, their allocation will be through opaque, sealed, sequentially numbered envelopes assigned to the interventions management team. 

The envelopes will be opened, in order, by the intervention team that will reveal the type of treatment to be assigned to the volunteers.

The study nature will prevent blinding of the intervention team and volunteers. However, the researcher that conducts the initial and final assessment of the outcomes and the statistician will be blinded to the allocation.

### 2.6. Intervention

The training protocol will last for 12 weeks (3 months) and comprise three times a week for 60 min each session. Before each session, the volunteers in arms G1 and G2 will be instructed by the intervention team to perform basic warming and stretching exercises accordingly to the Ramos protocol [21]. 

The ST consists of two sets (A and B) each to be performed alternately [39,40], as shown in Table 1, in accordance with recommendations of the American College of Sports Medicine (ACSM) for older adults [41] and Brazilian Diabetes Society (Sociedade Brasileira de Diabetes—SBD) [9]. A frequency of three days a week will be applied, very light to light intensity (40% to 50% of the test of a maximum repetition—1RM) and seven exercises involving the main muscle groups. In the 1RM test, a familiarization will be performed (pre-session), starting with 10% of body weight, progressing to the maximum lifting capacity, for a maximum of 5 attempts with an interval of 30 s between each attempt. The test will be performed bilaterally, starting with the nondominant side [42].

In addition, dumbbells and grounded muscle training equipment with different loads (Mega II, Marca Movement) will be used. The training will start initially with two sets of eight repetitions (1st to 4th week), progressing to two sets of twelve repetitions (5th to 8th week) and, finally, two sets of fifteen repetitions (9th to 12th week) [9,41]. The rest between sets will be sixty seconds and the load will gradually increase according to a good exercise performance. The strength training protocol will be applied to arms G1 and G2 equally.

Next, WBV will be applied through the Kikos P204–110v platform (Kikos Fitness, São Paulo-SP, Brasil), which has an oscillatory lateral vibration direction. The vibrating platform baseplate is slip-resistant for better safety during the training [7]. The amplitude (the vertical displacement of oscillatory motion) will be determined by the distance between both feet on the surface of the platform, ranging from approximately 2.0 mm to 5.0 mm [43]. A gradual increased in vibration frequency (the number of cycles of oscillation) and exposure duration was performed to avoid an early muscle activity decrease [44,45,46]. Vibration frequency and time spent between use and rest per week are described in Table 2.

On the platform, barefoot volunteers will adopt an isometric squat positioning at 40°, verified with a conventional goniometer. On their heels, silicon insole will be applied (semi-rigid) in order to keep only the forefoot and middle-foot in direct contact with the platform surface, thus correcting the body weight distribution and foot biomechanics changes. Both strategies will be used to minimize the axial transmission to cranial base [7,21,47]. In accordance with the International Organization for Standardization (ISO) 2631-1, all the parameters are considered safe for body exposure to mechanical vibration [48].

The WBV sham (simulation) will be performed with the platform unplugged. A speaker connected to the platform will emit a similar sound during the time equivalent to the protocol, since the vibrating stimulus cannot be visually differentiated. The volunteers that experience the simulation will have to adopt the same squat positioning during the session. We will assure no contact between volunteers from arms G1 and G2 [47].

The volunteers in arm G3 group will be instructed to manage their diabetes as usual routine without practicing physical activity. All of them will be informed that they will be scheduled to receive the treatment after 12 weeks. 

It is worth mentioning that the entire intervention team will be submitted to a complete 8 h training. 

### 2.7. Outcome Measurements

#### 2.7.1. Primary Outcomes


**Skin temperature**


The thermographic imaging of lower limbs will be captured by FLIR E40bx^®^ camera (FLIR^®^ Company, Wilsonville, Oregon, EUA), with thermal sensibility of 0.5 °C, light emission value of 0.987 and 160 × 120 pixels resolution. The camera will be positioned at a distance of 1 m from the regions of interest with a 90° angle (perpendicular) relative to the camera lens [49].

The volunteers will have fifteen minutes for acclimatization and rest as the body temperature comes into alignment with the room temperature. The test room temperature will be controlled at 18 °C and 23 °C with relative air humidity between 40 and 70% monitored with a precise thermo hygrometer (KT-908). All volunteers will be previously oriented to avoid pain killers or vasoactive drugs, lotions or moisturizing creams on the regions of interest, in addition to caffeine and nicotine in the 4 h before the exam [49].

After the acclimatization period, the thermal images of the interest points will be registered and, subsequently, the mean temperature will be analyzed by the thermographic software provided by the manufacturer (FLIR^®^ Tools). Table 3 shows the six regions of interest [50]. All the photographic images will be collected by one evaluator only with expertise in managing the camera.


**Mean peripheral arterial blood flow velocity**


Mean dorsal pedal and posterior tibial arteries blood flow velocity will be measured by a handheld Doppler ultrasound (RH-BV-620VP, Guangdong, China/Mainland) with continuous sinusoidal wave emission. The skin site for the test will be coated with a water-soluble gel, and the high-frequency (8.0 MHz ± 10%) transducer will be placed at a 45° angle in relation to the blood vessel analyzed (against the flow) [44]. For measurements related to the dorsal artery of the foot and posterior tibial artery, foot dorso and distal third of the calf will be the area of reference, respectively. The volunteers will be positioned seated with supported feet and the confirmation of the most adequate site will be performed based on the audible signals generated by the Doppler device. The measurements will be expressed in centimeter per second (cm/s), using the mean of three measures of each artery with a 60 s interval [51]. All examinations will be performed by the same experienced examiner.


**Mobility**


Functional mobility will be tested by Timed Up and Go test (TUG), which is a rapid test and sensible for detecting changes in older adults. It consists of standing up from an armless chair, walking 3 m away to a marked post, turning 180 degrees, and walking back to the chair and sitting down without support. The chair to be used has a seat height of 45 cm. Older adults without functional mobility deficit are able to complete the test in less than 10 s [52]. It will be performed three times with 30 s intervals and the result considered will be the mean of performance timed by a stopwatch (Oregon Scientific, Portland, OR, USA, model SL-210).

#### 2.7.2. Secondary Outcomes


**Quasi-static posture**


An optoelectronic system BTS SMART DX100 (BTS Bioengineering, Milan, Italy) with four cameras equipped with 850 nm LED illuminators sensible to infrared (resolution 640 × 480) will be used to measure body alignment from the lower limbs. Before capturing data, a calibration process (static and dynamic) will be performed for spatial accuracy in reconstructing the three-dimensional model from this horizontal (x and y) and vertical (z) coordinate system. The maximum error of spatial positioning reconstruction is 0.5 mm [53].

After collecting the anthropometric data (height, weight, distance between anterosuperior iliac spines, lower limbs length and pelvic cavity depth), twenty-two reflective markers (15 mm of diameter) will be attached with double sided-tape on anatomical reference sites in accordance with Davis model (Figure 2) [54].

All volunteers will be invited to stand in a marked spot, keeping a rest position with arms extended by their side for a 10 s registration period. After capturing and reconstructing the 3D model, data will be imported and processed by SMART-Clinic^®^ software. From the anatomical points referenced, the following will be evaluated: pelvic obliquity (PO), pelvic tilt (PT), pelvic rotation (PR), hip ad/abduction (HA), hip flexion/extension (HF), hip rotation (HR), knee valgus–varus (KV), knee flexion/extension (KF), knee rotation (KR), ankle plantar/dorsiflexion (AP) and foot rotation (FR) [54].


**Plantar pressure and body balance**


MPS Biomech^®^ Baropodometric Modular Platform (LorAn Engineering, Bolonha, Itália) with 50–100 Hz acquisition and compatible with software Studio BIOMECH^®^ (Letsense group) will be used to determine plantar pressure and body balance in open-eyed (OE) and close-eyed (CE) conditions.

The barefoot volunteers will be instructed to stand in the middle of the platform, with a loose and habitual posture, bipodal stance and arms extended along their body. The volunteer will be asked to keep eyes directed to the fixed point in front of them in order to control possible changes in the visual field under OE condition [55]. Before capturing data, a 30 s adaptation period will be allowed. Each record will last for 20 s and it will be invalidated in cases of the participant moving or unbalancing, which will require the acquisition process to be repeated. The mean of three measures will be used for data analysis.

The evaluation of plantar pressure, bilaterally, will consider maximum and mean pressures, contact area and mean percentage of weight distribution. To measure the balance, the following variables will be analyzed: center of pressure displacement (COP), COP velocity, anteroposterior displacement (X axis) and mediolateral displacement (Y axis).

### 2.8. Monitoring Measurements


**Vital signs and capillary glycemia**


Before and right after each session, systolic (SBP) and diastolic (DBP) blood pressure will be measured by an aneroid sphygmomanometer and a stethoscope (Premium, China), and will be classified according to the recommendations from the Brazilian Society of Cardiology [56].

The number of breaths per minute (RR) will be registered by the chest movement or abdominal compartment for a 60 s period, and the heart rate (FR) and peripheral oxygen saturation (SpO2) will be measured by a portable pulse oximeter (More Fitness, MF-416, China) [57]. 

Capillary glycemia will be monitored by a portable blood glucose meter (ACCU-CHEK^®^ Active, Roche Diabetes Care, model GU) in accordance with the parameters recommended by ISO 15197:2013 [58].


**Other biochemical markers**


In the initial assessment and in the reassessment after 12 weeks, glycated hemoglobin (HbA1c) and lipid-related cardiovascular risk factors (total cholesterol, triglycerides, high density lipoprotein—HDL, low density lipoprotein—LDL and very low density lipoprotein—VLDL) will be analyzed [20].

### 2.9. Data Processing and Analysis

All data records will be kept under custody of LACIRTEM lab and an allocation-blind statistician will conduct all analysis using Statistical Package for the Social Science (SPSS, IBM^®^, EUA), version 23.0, with 5% (*p* < 0.05) level of significance.

Sociodemographic, anthropometric and baseline clinical data for both groups will be expressed as the mean and standard deviation for continuous variables, proportions and percentages for nominal variables. The Kruskal–Wallis H test will be used for continuous variables and Chi-square test will be used to associate nominal variables in order to verify homogeneity between baseline data among the groups. 

All analysis will be conducted in accordance with the intention-to-treat method (ITT) using the last observation carried forward (LOCF) method, which inputs baseline data at postintervention for the dropout cases [59]. Missing values due to participants’ nonattendance or withdrawal are expected in clinical trials. Continuous variables with normal distribution will be expressed as mean and standard deviation and the intra-group comparison will be performed by paired *t*-test, and the inter-group comparison by analysis of variance (ANOVA). For nonnormal continuous variables, data will be expressed in median and interquartile intervals (Q25–Q75) and compared within the groups by Wilcoxon test and among the groups by Kruskal–Wallis H test. Size effect will be calculated to support the statistical significance. 

We will also investigate and register the possible adverse events during the treatment protocol. 

## 3. Discussion

This study presents an overview of a training program and a detailed description of the assessment methods in an older population with DM2. A properly prescribed and regularly performed physical activity is considered an essential tool for the management of DM2 and its complications [9]. However, low compliance rate to physical activity programs is one of the barriers faced by healthcare professionals, especially from the diabetic older population [60]. Therefore, new, effective, long-term and low-cost strategies to keep this population active are needed. WBV fits as a promising training modality that is feasible, practical and safe and, when combined with ST, can provide additional effects [61].

The assessment of vascular function is a primary outcome, aiming to reduce the risk of cardiovascular complications. The endothelium is essential to maintain vascular homeostasis; however, when dysfunctional, it loses its physiological properties and undergoes changes [18]. Thus, two important tools will be used, the infrared thermographic image (ITIV) and the portable vascular Doppler ultrasonography (UDV) for quantitative evaluation and mapping of these changes. ITIV is a useful method of assessment that allows the measurement and recording of the heat radiated by the skin surface, without providing ionizing radiation or physical contact [62], and UDV is a practical, relatively low-cost and sensitive method for assessing the mean blood flow in small vessels [63].

Functional mobility is also a primary outcome of this study, considering that important loss occurs during the aging process of people with DM2. Therefore, TUG is a simple and effective assessment method, for which high scores are associated with impaired balance, reduced gait speed and difficulty in activities of daily living [52].

Finally, if effective, this study will provide a nonpharmacological treatment option of innovative and scientific value.

## 4. Strengths and Limitations of the Study

The development of a protocol study for a single-blind, randomized, controlled clinical trial that is able to identify whether whole-body vibration (WBV) provides additional benefits to strength training (ST) in lower-limb blood flow and mobility in older adults with type 2 diabetes mellitus (DM2). The vibrating platform is safe equipment, easy to handle and low-cost, which minimizes the need for conscious effort; therefore, it is most suitable for obese and sedentary patients. This new tool may be a strategy for lower-limb blood flow assessment associated with infrared thermographic imaging analysis and mean peripheral arterial blood flow velocity evaluated by a handheld Doppler ultrasound to identify physiological changes. The limitations are an absence of clinical outcomes such as quality of life, and that the vibration sham (a device emitting a similar noise to WBV) can produce physiological effects due to static squatting.

## Figures and Tables

**Figure 1 diagnostics-12-01550-f001:**
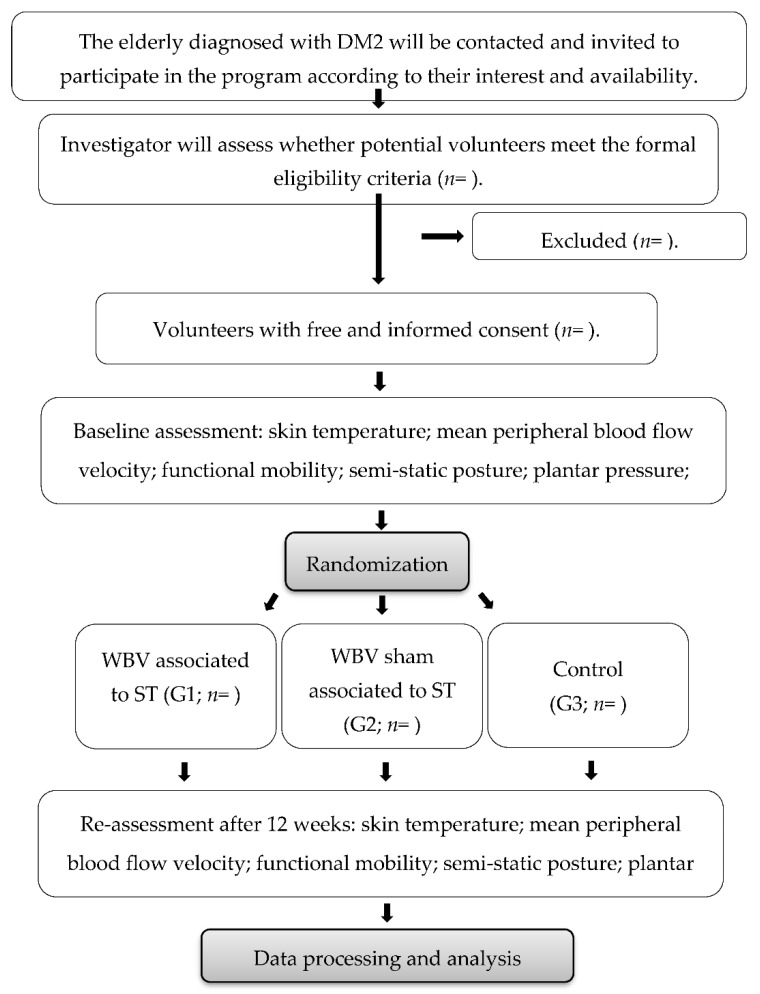
Enrollment, randomization and assessment flow chart. DM2, type 2 diabetes; WBV, whole-body vibration; ST, strength training.

**Figure 2 diagnostics-12-01550-f002:**
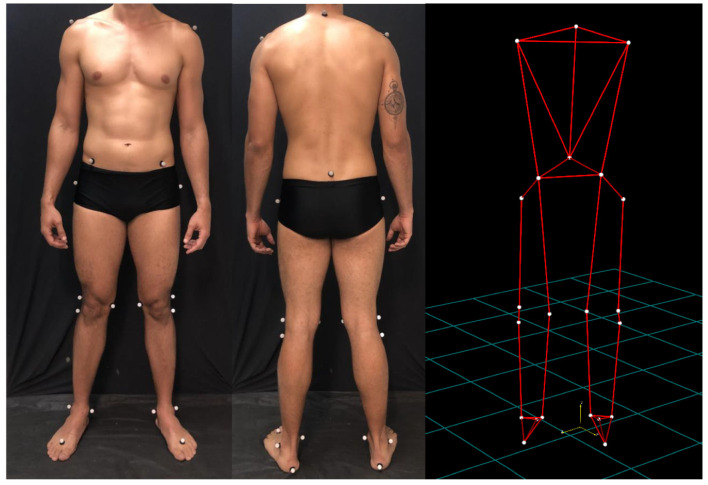
Placement of markers in the body the volunteer according to the Davis protocol.

**Table 1 diagnostics-12-01550-t001:** Strength training regimens.

Set A	Set B
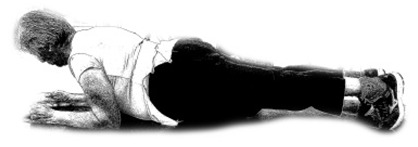 Prone Plank Muscles Worked: Abdominal rectum, external oblique, serratus anterior, pectoralis major, tensor fascia lata, quadriceps.	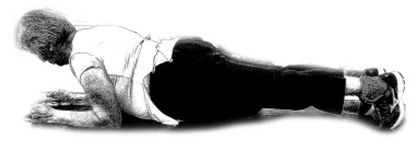 Prone Plank Muscles Worked: Abdominal rectum, external oblique, serratus anterior, pectoralis major, tensor fascia lata, quadriceps.
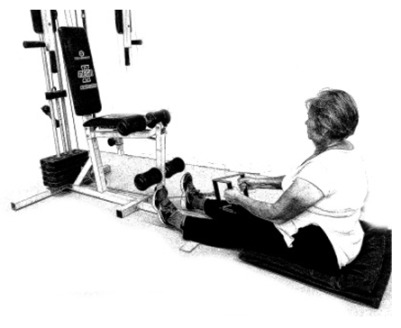 Seated Low Row Muscles Worked: Trapezius, latissimus dorsi, rhomboids, posterior deltoid.	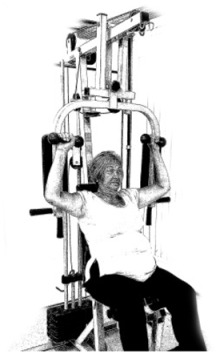 Dumbbell Shoulder Press Muscles Worked: Anterior deltoid, lateral deltoid, triceps, trapezius, upper chest.
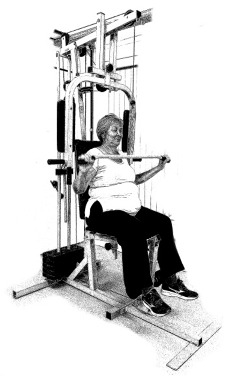 Lat Pulldown Machines Muscles Worked: Latissimus dorsum, posterior deltoid, lower trapezius, rhomboids.	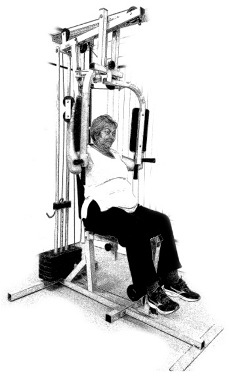 Chest Fly Machine Muscles Worked: Pectoralis major, anterior deltoid.
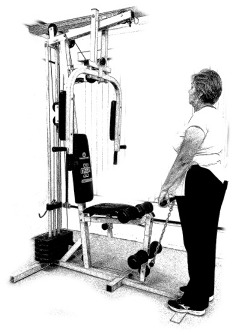 Upright Cable Row Muscles Worked: Lateral deltoid, trapezius, anterior deltoid, supraspinatus, infraspinatus, minor round.	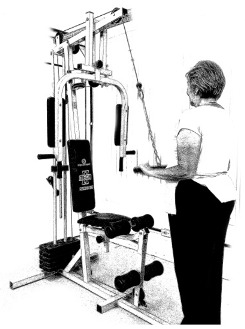 Triceps Pushdown Muscles Worked: Triceps, deltoid, forearm.
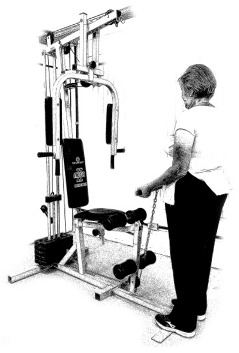 Standing Bicep Curl (Cable) Muscles Worked: Biceps, brachialis, brachioradialis, anterior deltoid, forearm.	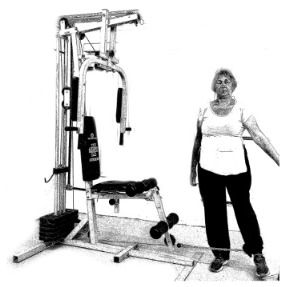 Cable Hip Abduction Muscles Worked: Gluteus maximus, gluteus minimus, tensor fascia lata.
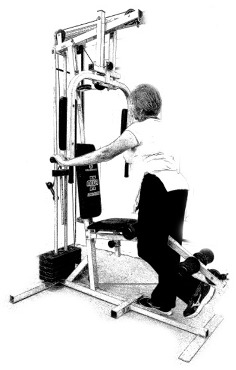 Standing Leg Curl Muscles Worked: Isquiotibiales, gluteos, sural triceps.	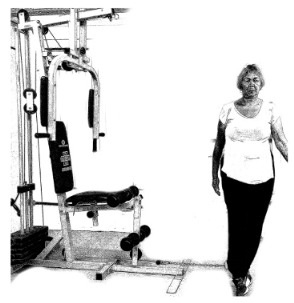 Cable Hip Adduction Muscles Worked: Long adductor, short adductor, magnum, gracilis, pectinous.
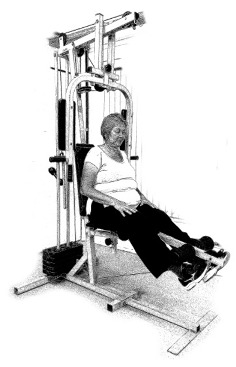 Leg Extension Muscles Worked: Quadriceps, anterior tibialis.	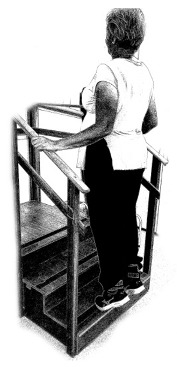 Standing Calf Raise Muscles Worked: Sural Triceps.

**Table 2 diagnostics-12-01550-t002:** Description of the whole-body vibration (WBV) protocol over 12 weeks.

Weeks	Time per Exercise (s)/Number of WBV Exercise	Frequency (Hz)	Rest Period (s)	WBV Total Time (s)
1–2	30/8	16	30	480
3–4	30/8	18	30	480
5–6	45/8	20	30	720
7–8	45/8	22	30	720
9–10	60/8	24	30	960
11–12	60/8	26	30	960

**Table 3 diagnostics-12-01550-t003:** Regions of interest (ROI) to be analyzed.

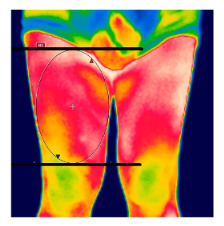	ROI: Anterior thigh region. Upper limit: groin. Lower limit: horizontal line aligned with the apex of the patella. Shape: ellipse, following the outline of the thigh.
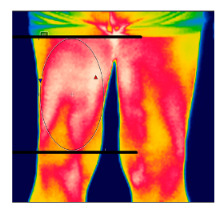	ROI: Posterior thigh region. Upper limit: horizontal line aligned with the line gluteal. Lower limit: horizontal line aligned with the tip of the fibula. Shape: ellipse, following the outline of the thigh.
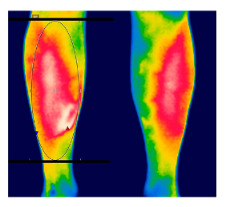	ROI: Anterior leg region. Upper limit: horizontal line aligned with the tip of the fibula. Lower limit: horizontal line above the malleolus. Shape: ellipse, following the outline of the lower leg.
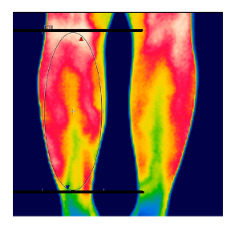	ROI: Posterior leg region. Upper limit: horizontal line aligned with the tip of the fibula. Lower limit: horizontal line above the malleolus. Shape: ellipse, following the outline of the lower leg.
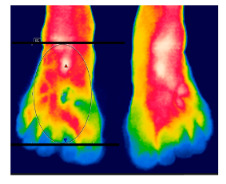	ROI: Dorsal foot region. Upper limit: horizontal line aligned with the talocrural joint. Lower limit: horizontal line aligned with the metatarsophalangeal joint. Shape: ellipse, following the outline of the foot.
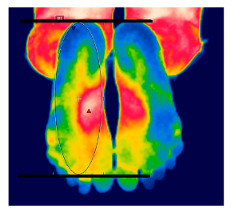	ROI: Plantar foot region. Upper limit: horizontal line aligned with the heel tip. Lower limit: horizontal line aligned with the metatarsophalangeal joint. Shape: ellipse, following the outline of the foot.

## Data Availability

The results of our original study will be disseminated after completion in peer-reviewed journals and presented at conferences.

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
