# Peer review of "Whole-Body Vibration Associated with Strength Training on the Lower-Limb Blood Flow and Mobility in Older Adults with Type 2 Diabetes: A Study Protocol for a Randomized Controlled Trial"

_diagnostics, 2022, doi:10.3390/diagnostics12071550_

Round 1
Reviewer 1 Report
The authors presented a research protocol dedicated to diabetics using whole body vibration strength training.
I have some comments that I think could be used by the authors to improve the study protocol.
1) Strength training should be supplemented with: intensity (%1RM), pace of movement (slow, medium, fast). Number of repetitions should be adjusted to %1RM. In addition, please specify what the training is aimed at (its goal?): hypertrophy? endurance? strength? muscle power? prevention of sarcopenia?
2) whole body vibration: a better characterization of the stimulus is needed: what about acceleration? constant or variable frequency?
Please justify the increase of frequency in the following workouts.
3) add to the exclusion criteria all conditions related to fall risk and balance disorders.
4) If the intervention is related to diabetes mellitus biochemical parameters should be added to the protocol: e.g. glucose, glycated haemoglobin, insulin, glucagon, nitric oxide, vasodilatory factors, vascular stiffness (PWA, PWV), inflammation (interleukins) etc.
5. the authors have omitted several recent papers, the results of which should be taken into account in protocol planning:
(a) microcirculation: Piotrowska et al. 2022 Changes in Skin Microcirculation Resulting From Vibration therapy in Women with Cellulite IJERPH, 19, 3385 https://doi.org/10.3390/ijerph19063385
(b) Glucose consumption: Maciejczyk et al. Acute Effects of Whole-Body Vibration on Resting Metabolic Rate and Substrate Utilisation in Healthy Women. Biology, 11(5), 655.
(c) Resistance training: Rosenberger, A., et al. (2019). Changes in motor unit activity and respiratory oxygen uptake during 6 weeks of progressive whole-body vibration combined with progressive, high intensity resistance training. Journal of Musculoskeletal & Neuronal Interactions, 19(2), 159.
(d) vibration characteristics: Kang et al. Metabolic responses to whole-body vibration: Effect of frequency and amplitude. Eur. J. Appl. Physiol. 2016, 116, 1829-1839.
e) Rittweger, J. Metabolic Responses to Whole-Body Vibration Exercise. In Manual of Vibration Exercise and Vibration Therapy; Springer: Cham, Switzerland, 2020;
Author Response
Dear Editor,
We appreciate the kind consideration of this paper by the reviewer and the opportunity to highlight the value of our research. The paper has been improved as detailed below, based on reviewer’s thoughtful comments. We have noted our changes in “red” so they are easy to locate. We thank you sincerely for your time and consideration.

Reviewer 2 Report
Dear Authors
The title of this study seems to be consistent with the Diagnostics. Title is “Whole-Body Vibration Associated with Strength Training on the Lower-Limb Blood Flow and Mobility in Older Adults with Type 2 Diabetes: a Study Protocol for a Randomized Controlled Trial” I think this paper will be better if some minor and major points are corrected.
Minor points
1. Ethical content (lines 42 to 44) need not be presented in the abstract.
2. All statistic symbol 'p' must be written in italics. The same should be done in all cases presented below.
3. The 'n' for the subject should be changed to italics.
4. In line 142, the '-' representing the range of BMI is replaced with an 'a'. Please correct it. The comma would be changed a decimal point. And you must change the '2' in 'kg/m2' to the square (kg/m²). For example, (25,0 a 29,9 kg/m2) → (25.0 – 29.9 kg/m²)
5. Please correct all in the format of MDPI papers.
Major points
1. This study seems to have well-organized experimental procedures and methods, and the overall content of the article is very interesting. However, if the subjects of the study will be type 2 diabetes patients, it would be good to include blood sugar and HbA1c levels in the variables to be measured.
2. A full explanation of what low frequency WBV (16-26 Hz) means should be given. In addition, the meaning of the range of vibration of WBV, the intensity of vibration, and the frequency of vibration should be presented.
3. The amount of discussion is rather small compared to the overall experimental content. If the volume is small because there are no results of the study, it is also necessary to explain the meaning of each variable that the authors want to see. And, there are many papers related to WBV so far. Readers will be able to better understand if these papers also include the study of WBV treatment in elderly and diabetic patients.
Sincerely,
Author Response

(The authors gave the same response as above.)

Round 2
Reviewer 1 Report
The authors submitted a second version of the manuscript. In my opinion, they failed in revising the manuscript and did not incorporate my suggestions.
If the authors chose a group of patients with diabetes, they should have been aware that it is necessary to control the effects of the intervention on the health status of the subjects and the assessment of biochemical markers associated with diabetes. Lack of funding is not an excuse. The research protocol should be correct and the authors should try to secure funding. Especially since the protocol can be used by scientists from other countries and such presented protocol, without biochemical tests, can be misinterpreted and replicated. In my opinion, it is necessary to include analysis of blood biochemical markers related to diabetes. Measurement of blood glucose alone is insufficient and somewhat misses the point because it reflects only temporary changes in blood glucose.
2. the planned workout (training) should be described in detail so that it can be assessed whether it is properly planned; a reference to recommendations is not sufficient
3. the paper still does not take into account the latest relevant references
Author Response
RESPONSE TO THE REVIEWERS
Dear Reviewer,
We appreciate the kind consideration of this paper by the reviewer and the opportunity to highlight the value of our research. The paper has been improved as detailed below, based on reviewer’s thoughtful comments. We have noted our changes in “red” so they are easy to locate. We thank you sincerely for your time and consideration.
- Reviewer #1 Thank you for revising the manuscript. The issues are corrected as follows:
- Lack of funding is not an excuse. The research protocol should be correct and the authors should try to secure funding. In my opinion, it is necessary to include analysis of blood biochemical markers related to diabetes. Measurement of blood glucose alone is insufficient and somewhat misses the point because it reflects only temporary changes in blood glucose.
Thank you, this is an important consideration. Second subtitle of item 2.8, page 13. However, at the moment, the biological markers integrated into the protocol submitted and validated by our institution (CEP/CCS/UFPE) are the ones that were integrated into the article. But we will consider the other markers and make a addendum to the ethics committee to include in the finished article.
- The planned workout (training) should be described in detail so that it can be assessed whether it is properly planned; a reference to recommendations is not sufficient.
Thank you very much for this suggestion. It is now updated in page 5 and 6, second and third paragraph of the Intervention (Ref. 6).
- The paper still does not take into account the latest relevant references
Thank you very much for this suggestion. References number 22, 23 and 46 were added according to your suggestion and they are extremely important for the development of the research.
Reviewer 2 Report
The authors responded to all comments.
Nothing to add.
Author Response
RESPONSE TO THE REVIEWERS
Dear Reviewer,
We appreciate the kind consideration of this paper by the reviewer and the opportunity to highlight the value of our research. The paper has been improved as detailed below, based on reviewer’s thoughtful comments. We have noted our changes in “red” so they are easy to locate. We thank you sincerely for your time and consideration.
- Reviewer #2 Thank you for revising the manuscript.
The authors responded to all comments. Nothing to add.
Thank you.
Round 3
Reviewer 1 Report
I encourage the Authors to expand biochemical testing, in diabetics this may be especially important. The inclusion of glycated hemoglobin and lipid indices is satisfactory. Thank you. Well done!